# Phenological Mapping of Invasive Insects: Decision Support for Surveillance and Management

**DOI:** 10.3390/insects15010006

**Published:** 2023-12-22

**Authors:** Brittany S. Barker, Leonard Coop

**Affiliations:** 1Oregon Integrated Pest Management Center, Oregon State University, 4575 Research Way, Corvallis, OR 97333, USA; coopl@oregonstate.edu; 2Department of Horticulture, Oregon State University, 4017 Agriculture and Life Sciences Building, Corvallis, OR 97333, USA

**Keywords:** degree-day model, forecast, pest, monitoring, gridded climate data, spatial phenology model

## Abstract

**Simple Summary:**

Phenological maps can depict the development and seasonal activities (phenology) of invasive insects at area-wide scales, such as counties, states, or entire nations. When regularly updated using real-time and forecast climate data, these maps may improve the timeliness of early detection and control tactics that target specific life stages. Rapid responses to invasive insects may increase the likelihood that populations are eradicated or controlled before they can spread or increase in size. In this review, we provide a brief history of phenological mapping, compare three types of maps that are commonly used for real-time decision support, and summarize climate datasets that may be used for mapping. We also present applications of phenological maps for assessing establishment risk, investigating pest–host interactions, and measuring climate-driven changes in pest phenology. Next, we discuss model complexity, potential sources of model error and uncertainty, methods for evaluating map predictions, and recommendations for future research. The development of additional real-time climate datasets and pest models will allow expanded use of phenological maps to help control invasive insects under current and future climates.

**Abstract:**

Readily accessible and easily understood forecasts of the phenology of invasive insects have the potential to support and improve strategic and tactical decisions for insect surveillance and management. However, most phenological modeling tools developed to date are site-based, meaning that they use data from a weather station to produce forecasts for that single site. Spatial forecasts of phenology, or phenological maps, are more useful for decision-making at area-wide scales, such as counties, states, or entire nations. In this review, we provide a brief history on the development of phenological mapping technologies with a focus on degree-day models and their use as decision support tools for invasive insect species. We compare three different types of phenological maps and provide examples using outputs of web-based platforms that are presently available for real-time mapping of invasive insects for the contiguous United States. Next, we summarize sources of climate data available for real-time mapping, applications of phenological maps, strategies for balancing model complexity and simplicity, data sources and methods for validating spatial phenology models, and potential sources of model error and uncertainty. Lastly, we make suggestions for future research that may improve the quality and utility of phenological maps for invasive insects.

## 1. Introduction

The protection of agricultural and natural resources depends on the precise timing of surveillance, monitoring, and management of invasive insect populations [1,2,3]. Scheduling pest control tactics according to calendar dates and expectations of the “normal” time in which seasonal activities (phenology) of pests have occurred in previous years is often ineffective because rates of insect development often vary annually due to variations in weather [4,5,6,7]. Conversely, modeling the phenology of an insect species using information on their life cycle requirements and climate data for the current year may increase the precision of estimates of dates when important seasonal events occur, such as the first adult emergence and egg hatch [4,7,8]. This information may improve the effectiveness and cost efficiency of early detection and management programs because these programs often target life stages that are more observable (e.g., larvae vs. adults of wood-boring beetles) or more vulnerable to control tactics, such as pesticide treatments [3,9,10,11,12,13].

Degree-day models are widely used in decision support systems that predict the phenology of agricultural insect pests because of their simplicity and ability to accommodate multiple species with varying life histories [8,9,10,14,15,16,17,18,19,20]. The development and phenology of an organism in a degree-day model is driven by heat accumulation above a lower temperature threshold (and oftentimes below an upper temperature threshold) over a daily or weekly time step (Figure 1) [4,5,7,21,22,23,24]. The lower developmental temperature is often referred to as the base temperature. Typical examples of degree-day models that have been used for many years include those that predict first egg hatch of the codling moth [*Cydia pomonella* (L.)] in tree fruits [13,25,26,27,28], first emergence of the western cherry fruit fly (*Rhagoletis indifferens* Curran) [29], and adult flight, egg hatch, and larval development of the spongy (formerly “gypsy”) moth [*Lymantria dispar* (L.)] [30,31,32,33,34].

Most degree-day models for invasive insect species use climate data for a specific site, such as a weather station. Site-based model predictions, such as degree-day accumulations and dates of phenological events (e.g., first adult flight), are typically displayed in tabular and/or graphical formats (Figure 1A). This information is useful for decision support for pest surveillance or management in small areas, such as a forest parcel, fruit orchard, or vineyard [3,9,10,13,35]. However, site-based models are less applicable for decision support at area-wide scales, such as large counties or states, because phenology may vary spatially due to geographic factors (e.g., latitude, elevation, and continental effects), anthropogenic disturbances, and biological factors, such as spatial variations in population development and host plant availability.

Model predictions for multiple sites (e.g., in the form of number values, color-coded pins, or other expressions) can be overlayed on a base map to visualize geographic variation in insect phenology [8]. For example, degree accumulations for each site can be shown as text labels on an elevation map, which can be referenced to a key of degree-day requirements for important phenological events (Figure 1B). Nonetheless, maps with discontinuous model predictions may be of limited use for decision-makers in areas that lack model predictions, particularly for topographically and climatically complex areas where phenology can vary over short distances. 

This review focuses on the continuous mapping of phenology (hereafter phenological mapping), in which the phenology of a species is modeled over an entire geographic area (Figure 1C). The development of computer methods to relate phenological and meteorological observations to geography beginning in the 1970s significantly advanced the field of phenological mapping [36,37]. The first known phenology maps used for integrated pest management (IPM) decision support were for the codling moth in Michigan [38]. These SYMAP-generated maps depicted the predominant stages of the codling moth (eggs, larvae, pupae, and adults) over two generations for all dates during the growing season. Phenological mapping of IPM pests became more common beginning in the 1990s with further advancements in computers and a growing number of digitized geographic datasets and geographic information systems (GIS) software options [32,39,40,41,42].

Digital spatial climate datasets developed over the past ca. 30 years have increased the robustness and timeliness of phenological maps. Climate data at fine-scale spatial resolution increased the prediction accuracy because local effects (e.g., of mountains, valleys, or large bodies of water) that might influence phenological events could be modeled [43,44,45]. Additionally, advancements in satellite, cellular, and internet communications allowed meteorological observations to be released within hours of being collected, enabling researchers to produce phenology maps in near real-time to gain insight into the current development and activities of a species [15,16,46,47]. For example, the advent of free, near real-time (henceforth real-time for brevity) daily climate data from the Parameter-elevation Regressions on Independent Slopes Model (PRISM) database [http://www.prism.oregonstate.edu (accessed on 19 December 2023)] in 2011 was a major breakthrough for phenological mapping for the contiguous United States (CONUS), as PRISM data prior to this time were only available as monthly grids that were released months after data collection [43,44]. 

## 2. Data Requirements for Degree-Day Models

Detailed reviews on data requirements and methods for degree-day modeling of insects already exist [17,23,24,48,49] and are, therefore, only summarized here. Degree-day models are often developed using experimentally collected data on temperature–development relationships to estimate parameters, such as developmental rates, developmental temperature thresholds, duration of life stages, and stage-specific events [17,50,51,52]. These data are combined with daily temperature data, typically minimum and maximum temperatures (*T_min_* and *T_max_*, respectively), to estimate degree-days using various calculation methods [4,7,14,24,50,53]. A start date or biological event, such as the first flight, is needed to synchronize the insect phenology model to field populations [24,54,55].

## 3. Types of Phenological (Degree-Day) Maps

Below, we review and discuss some advantages and disadvantages of three types of maps that are often produced by spatial phenology models: (1) generic degree-day maps, (2) degree-day lookup table maps, and (3) phenological event maps. Maps produced by web-based platforms used for real-time decision support for detecting and controlling invasive insects for CONUS (Table 1) are used as examples. 

### 3.1. Generic Degree-Day Map

Generic degree-day maps show the current degree-day accumulations based on one or more standard lower temperature thresholds. For example, degree-day maps at Michigan State University’s Enviroweather [https://www.enviroweather.msu.edu (accessed on 19 December 2023)] use a standard lower temperature threshold of 50 °F (10 °C), and those at USPest.org [https://uspest.org/wea/index.html#DDMAPS/ (accessed on 19 December 2023)] use thresholds of 32, 41, or 50 °F (0, 5, or 10 °C, respectively; Figure 2). Degree-day maps at the University of Wisconsin’s AgWeather Vegetable Disease & Insect Forecasting Network [https://agweather.cals.wisc.edu/vdifn?p=insect (accessed on 19 December 2023)] use several different species-specific thresholds. 

Generic degree-day maps provide a good, general reference to show how the season is progressing, especially when compared to averages, such as 30-year normal degree-day maps. However, predictions of current degree-day accumulations are not matched up with insect life stages, so they must be used with experience and care for guiding surveying and management activities. For example, the Russo et al. [33] spongy moth egg hatch model could be used with a generic base 3 °C degree-day map. Once the map shows that the egg hatch degree-day requirement is approaching (317 degree-days), appropriate management activities could be initiated.

Enviroweather’s degree-day models use 1 March as the start date and includes only Michigan and Wisconsin maps, whereas models at USPest.org at https://uspest.org/ (accessed on 20 December 2023) uses 1 January as the start date and covers all of CONUS with separate mapping of major subregions (e.g., Midwest, Northwest, Southeast) and of states in the Pacific Northwest. Both websites are updated daily to produce maps based on climate data for the current year and historical averages (30-year normals). An additional map depicts the difference between historical and current year degree-day accumulations. AgWeather’s degree-day maps include either Wisconsin or the entire upper Midwest, and they allow end users to change the default model start date for individual species.

The degree-day mapping program at USPest.org [https://uspest.org/cgi-bin/usmapmaker.pl (accessed on 19 December 2023)] first appeared online in 1998 as a decision support tool for pest management in Oregon. The mapping region was expanded to include the entire Pacific Northwest region by 2002, and then to include CONUS at a higher spatial resolution (800 m) by 2005 [56,57]. Degree-days based on 30-year normals (centered on 1995) are calculated using gridded monthly temperature data from the PRISM database, while daily real-time degree-days are calculated using data from thousands of weather stations in the USPest.org collection of public networks. For the map creation, the degree-day mapping program uses a ‘climatologically aided interpolation’ method (sometimes more generally referred to as ‘delta correction’) that uses a gridded climate dataset, such as PRISM, to improve the interpolation of a site-based dataset, such as recent station observations (Figure 2A) [45,58]. More detail on the processes involved in the map production is documented at https://uspest.org/wea/mapmkrdoc.html (accessed on 19 December 2023).

Constructed in 2017, a second custom online phenology mapping program at USPest.org [https://uspest.org/dd/mapper (accessed on 19 December 2023)] was developed to offer an alternative and simpler workflow that uses real-time daily PRISM temperature grids and does not require data from multiple weather stations for correction (Figure 2B). However, the input data has a lower resolution (4 km), which, although adequate for most state-level maps, would be insufficient for small states or single growing regions that are topographically complex, such as Hood River County, Oregon. To address this issue, this version of the degree-day mapping program includes an option to downscale resulting degree-day maps to 800 m (Figure 2C) using a custom distance–elevation weighted regression algorithm, which is written in GRASS GIS and documented at https://uspest.org/dscale (accessed on 19 December 2023). The main map making program is written in R. An example of the R code used to calculate degree-days from PRISM daily climate data is provided in Appendix A.

### 3.2. Degree-Day Lookup Table Map

Degree-day lookup table maps show the current life stages or phenological events of an organism that correspond to specified values or ranges of accumulated degree-days for a specified date (Figure 3). Thus, degree-day accumulations, which are depicted in generic degree-day maps, are matched to specific points or events during the life cycle. 

For insects, life cycle points (and events) could typically include the egg stage present, egg hatch, larval stage present, pupal stage present, adult emergence and presence, and egg laying. The simplicity of the approach and its applicability to multiple organisms has sustained its use for several years. For example, the Degree-Day, establishment Risk, and Phenological event maps (DDRP) platform [16] at USPest.org [https://uspest.org/CAPS (accessed on 19 December 2023)], the USA National Phenology Network [https://www.usanpn.org/data/forecasts (accessed on 19 December 2023)], and SAFARIS (Spatial Analytic Framework for Advanced Risk Information Systems) PestCAST and Field Operations (FO) Weekly [https://safaris.cipm.info (accessed on 19 December 2023)] provide degree-day lookup table maps for several invasive insect species (Table 1) [15].

Specific features that make the degree-day lookup table approach so common include:A relatively straightforward workflow. The workflow of generating a degree-day lookup table map involves using gridded daily *T_min_* and *T_max_* data to calculate degree-day accumulations between a start date (usually 1 January, although some models use other start dates, such as 1 March) and a specified end date. Degree-day lookup tables are then used to associate degree-day accumulations with life stages, and output maps depict the results with color tables and legends.The use of common base thresholds for multiple species. As degree-day lookup table maps are relatively simple and generic, there is the potential to use the same lower temperature threshold base maps for multiple species. This contrasts with more complex models that would require a separate base map for each case because of the application of different parameter values, including lower and upper thresholds, calculation methods, start dates, or diapause. The use of upper thresholds is relatively rare, at least for degree-day lookup table maps. For example, the SAFARIS FO Weekly maps produced by models for the old world bollworm [*Helicoverpa armigera* (Hübner)] and brown marmorated stink bug [*Halyomorpha halys* (Stål)] are constructed using the same 54 °F base maps, with no upper threshold. At least three other species in the FO Weekly map series share the same map.An ability to provide a “snapshot in time” for a single date. This allows, for example, regular updates that provide a gradually changing view of the current or near-future status of insect phenology. For example, degree-day lookup table maps produced by DDRP every 2−3 days depict the life stage and generation of insects on the map issue date. The USA National Phenology Network’s Pheno Forecast maps take advantage of 7-day National Digital Forecast Database (NDFD) forecasts to provide a 1-week “look ahead” prediction for CONUS [47]. SAFARIS PestCAST maps include a 1-month forecast using a 7-day NDFD forecast followed by three weeks of recent 20-year average PRISM data [15].Relatively simple design requirements. Degree-day lookup table maps can be designed as very simple visualization tools, such as by designing legend items to display only the stage or activity of interest (e.g., adult flight or egg hatch). Other stages and activities can then be represented as merged entries. The practice of focusing end users on a single target event represents a clear trade-off in reducing complexity (of multiple life stages) for users who may need clear directions in implementing surveillance or management actions.

The degree-day lookup table approach, whether used for sites or for mapping, can be considered a “metamodel” or “abstract of a model,” which is essentially a simplified version of a more complex population model. For example, the USPest.org version [https://uspest.org/dd/model_app?spp=gm3 (accessed on 19 December 2023)] of the Sheehan spongy moth model [60] is actually a degree-day lookup table or “metamodel” of results of this more complex single species model, as implemented by the University of Wisconsin [https://agweather.cals.wisc.edu/thermal_models/spongy-moth (accessed on 19 December 2023)].

### 3.3. Phenological Event Map

In contrast to a degree-day lookup table map, a phenological event map depicts the dates on which accumulating degree-days reach a value (target degree-day total) that corresponds with a selected phenological event [16,32,40,61]. Phenological event maps may offer the following advantages over degree-day lookup table maps to support monitoring and surveillance programs for invasive insects:Standardization. Mapping dates of phenological events allows for the standardization of legends and color tables across multiple species and events. For example, the colors assigned to each range of dates in the legend (e.g., 1–8 January = dark blue, 9–16 January = medium blue, etc.) can be applied to several events within a species, as well the same or different events in other species. For example, a prototype phenological event map developed for the codling moth in Oregon in the 1990s [62] depicts dates of egg hatch using uniquely colored two- or three-day intervals that span 17 May to 11 June (Figure 4). An alternative approach is to depict dates relative to the map issue date (Figure 5). For example, Pheno Forecast maps produce by the USA National Phenology Network use a single target event, such as “adult emergence”, while earlier events are ignored or replaced by approximations of time to the target event, such as “adults expected in 1–2 weeks” and “declining activity”.
Figure 4Phenological event map predicting dates of egg hatch for the codling moth [*Cydia pomonella* (L.)] for Hood River, Oregon. The map was used to illustrate use of GIS and phenological modeling for the first U.S. Department of Agriculture supported “Areawide IPM Project For Pome Fruit Crops in the Western US” [62] but has not been previously published.
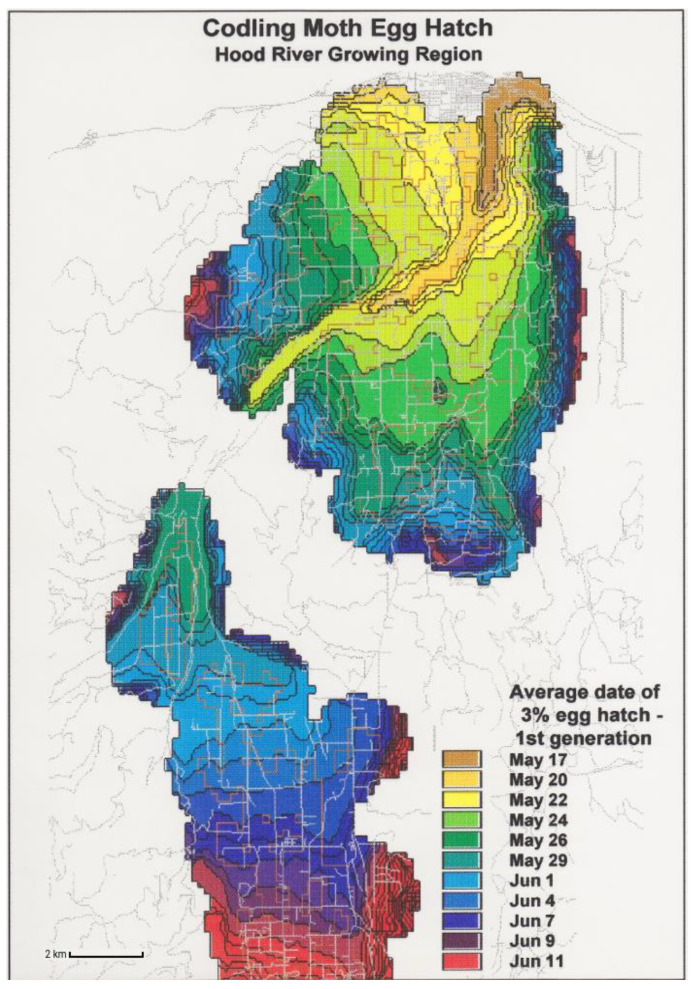

Figure 5Pheno Forecast produced by the USA National Phenology Network (USA NPN) for emerald ash borer (*Agrilus planipennis* Fairmaire) [63]. The map depicts the time to first adult emergence of the emerald ash borer in the contiguous United States relative to the map issue date (13 April 2023). Maps are updated every two days, which allows decision-makers to stay up to date on when this phenological event is expected. Reproduced with permission from the USA NPN, Emerald Ash Borer Adult Forecast; published by the USA NPN, 2023. Available online: https://www.usanpn.org/data/forecasts/EAB (accessed on 19 December 2023).
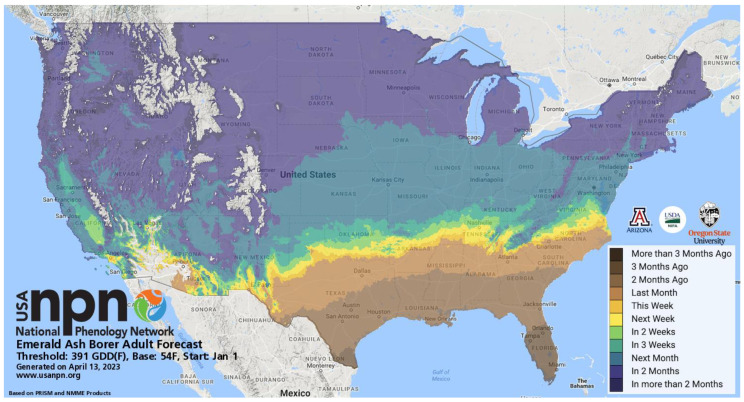

Operationally ready. Phenological event maps could be considered a more operational (tactical) product than degree-day lookup tables because they predict dates of events for a particular life stage, potentially up to weeks or months into the future. For example, a decision-maker may want to start planning their trap-setting operations several weeks before the estimated date of the first spring flight. Phenological event maps are more straightforward to interpret than degree-day lookup table maps because they do not require a mental conversion from stages into dates of events, which may reduce the learning curve for their use in decision support and allow for more direct communication of operational support. For example, phenological event map predictions could be merged with a calendar scheduling of monitoring or management activities, such as the dates of trap placement and removal. Nonetheless, operational readiness is presumptive, as most phenological event maps, at least for most invasive species, have not been formally tested in actual field use at this time.Simpler comparisons and expression of error rates. Phenological event maps allow for more direct comparisons of year-to-year variations of events than generic degree-day and degree-day lookup table maps. It is a relatively simple recordkeeping and reporting exercise to express differences in dates in the form of days difference [12,27,28]. This approach can also be used to express errors between predicted and observed events as discussed below (“8. Model validation”).

The prototype phenological event map developed for the codling moth in Oregon in the 1990s [62] (Figure 4) led to the development of platforms that were partially or fully automated and capable of generating phenological event maps for multiple insect species, including pests and beneficials. For example, Grevstad and Coop [61] developed partially automated phenological event maps that predicted the date at which *Galerucella calmariensis* (L.), a biological control beetle for purple loosestrife *(Lythrum salicaria* L.), would reach the photosensitive life stage across CONUS. Map predictions provided insight into whether the beetle would complete its life cycle and successfully overwinter at a given location, which may help guide decisions on where to release beetles.

The phenological event mapping approach of Grevstad and Coop [61] was improved upon further research and integrated into the DDRP platform [16]. As an example, Figure 6 shows a phenological event map produced by DDRP that depicts the average date of egg laying by the overwintering generation of the light brown apple moth [*Epiphyas postvittana* (Walker)] for 2023.

## 4. Applications of Phenological Maps

Phenological maps produced using real-time and forecast climate data have the potential to support the early detection of invasive pests. For example, regularly updated phenological maps on the SAFARIS platform provide decision support for the Cooperative Agricultural Pest Survey (CAPS) program [15,16], which conducts national and statewide surveys for exotic plant pests in the United States deemed to be of regulatory significance to the United States Department of Agriculture (USDA) Animal and Plant Health Inspection Service’s (APHIS) Plant Protection and Quarantine (PPQ) program [64]. SAFARIS models are developed for species on the USDA APHIS PPQ’s “National Priority Pest List” at http://caps.ceris.purdue.edu/approved-methods (accessed on 19 December 2023), which is updated annually to potentially add or remove pests. 

Phenological maps can support pest managers in timing treatments or other control tactics that target certain life stages. For example, phenological maps of egg hatch and larval development for the spongy moth were developed to support the timing of insecticidal sprays conducted for “stop the spread” programs in the eastern United States [30,31,32]. Similarly, pest models in the Pheno Forecast series were developed primarily for stakeholders who requested decision support for timing pest treatments [47]. Phenological maps also have the potential to help with scheduling the release of biological control agents, such as parasitoids, which typically target specific life stages in their insect hosts. 

At present, degree-day lookup table maps and phenological event maps are produced in real time for 33 invasive insect species for CONUS (Table 2). These include 16 models in the DDRP series, 21 models in the SAFARIS FO Weekly series, seven models in the SAFARIS PestCast series, and five models in the Pheno Forecast series. 

Of the 33 modeled species, 15 are established in CONUS, whereas the remaining 18 are at a high risk of establishing in this region (Table 2). DDRP and SAFARIS outputs are available in both static (image) and gridded (raster) formats. Pheno Forecasts are available in static formats and can be zoomed, queried, and panned using a visualization tool at https://www.usanpn.org/data/visualizations (accessed on 19 December 2023). Additionally, end users can sign up to receive email notifications that provide advanced warnings of when events will occur in their area.

Maps that depict a pest’s potential number of generations per year (i.e., voltinism) may help identify areas at risk of establishment because persistence in a new area requires a life cycle completion [15,65,66,67]. Additionally, maps of voltinism provide insight into expected levels of pest growth and subsequent damages to host plants for a given year [65,68,69]. Risk maps produced by SAFARIS identify potentially suitable areas based on survival-limiting temperatures as well as whether enough degree-days have accumulated for a species to complete a single generation or specified developmental stages [15]. DDRP produces generation maps but only considers climate stresses for establishment risk mapping [16]. 

Phenological maps may be used to assess the potential impacts of climate change on invasive insect phenology. For example, Barker et al. [46] tested for trends in the date of the first adult emergence of the emerald ash borer for North America and Europe after combining model outputs for each year over a recent 20-year period. Other studies have modeled the phenology of invasive insects under future climate change scenarios to estimate the impacts of global warming and altered precipitation patterns on important phenological events, potential voltinism, and population growth [42,68,69,70,71,72]. 

Mapping the extent to which critical time periods of insect life cycles coincide with phenological windows of host plant suitability may provide insight into pest establishment risk and outbreaks dynamics [34,73,74,75,76]. For example, Foster et al. [34] compared maps of Moderate Resolution Imaging Spectroradiometer (MODIS)-derived leaf out with maps of egg hatch for the spongy moth to identify areas that were most susceptible to defoliation in a given year. Simulations of the timing, locations, and severity of outbreaks in migratory insect pests have also included phenological mapping of pests and their host plants. For example, generic degree-days maps for the fall armyworm [*Spodoptera frugiperda* (J.E. Smith)] and corn plants were combined with an atmospheric dispersion model to predict the timing and direction of the multigenerational migration of this pest in CONUS [76,77].

## 5. Gridded Climate Data

Table 3 summarizes several gridded daily *T_min_* and *T_max_* datasets that may be suitable for phenological mapping, although it is not an exhaustive list of all available datasets. Specifically, we only report datasets that meet the following characteristics at the present time: contain observations for years up to at least 2016, cover large areas (global, regional, or country-wide scales), have a spatial resolution of at least 0.1° (ca. 11.1 km at the equator), and are publicly accessible.

Maps used for within-season decision support of invasive insects depend on having access to real-time daily *T_min_* and *T_max_* data with spatial resolutions that are appropriate for the needs of decision-makers. For example, phenological maps at a 4 km resolution are generally sufficient to support pest surveillance programs for the entire CONUS [15], but are probably not appropriate for smaller scales, such as a county or city. Real-time PRISM data with a spatial resolution of 4 km are freely available, and higher resolution (800 m) data can be purchased from the PRISM group. Real-time DDRP forecasts at USPest.org are produced using PRISM data (4 km resolution) as climatic inputs, whereas monthly updated North America Multi-Model Ensemble (NMME) 7-month forecasts or recent 10-year average PRISM data (calculated on a bimonthly basis) are used to predict pest phenology up to the end of the year [16].

Phenological mapping for within-season decision support in areas outside of the United States is typically hindered by a lack of real-time gridded daily *T_min_* and *T_max_* data (Table 3). However, historical datasets may be used for model development and validation, such as those for Europe [83], continental North America and Hawaii [84,85], Brazil [79,86], China [80,81], India [82], and Bangladesh, Nepal, and Pakistan [78]. For example, the validation of DDRP models for the emerald ash borer and the small tomato borer [*Neoleucinodes elegantalis* (Guenée)] used E-OBS and BR-DWGD data for Brazil and Europe, respectively [16,46]. 

Some phenological mapping studies overcame an absence of readily available gridded daily climate data by interpolating weather station data over a landscape of interest using custom software [31,40,41,87,88,89,90]. For example, the GEO-BUG platform offered four automated interpolation methods to map the date at which a pest insect species reached a specified life stage in the United Kingdom [41,88]. Interpolation methods commonly applied to *T_min_* and *T_max_* estimates include those based on distance analyses (e.g., inverse distance weighted and spline interpolation) or geostatistics (e.g., kriging and multiple regression) [33,58,88,89,90,91]. To our knowledge, however, there are no presently available platforms that use interpolation methods to produce real-time phenological maps for insect pests. 

## 6. Potential Sources of Error and Uncertainty

Common sources of error in insect phenology models include natural population variability, microclimatic factors, anthropogenic disturbances (e.g., land use patterns), and biotic factors (e.g., migration, host quality, competition, predation, and disease) [17,23,40,48,52,87,88]. We refer readers to Chuine and Régnière [23] and Coop and Barker [17] for detailed reviews of this topic. Additionally, most invasive insect species are poorly studied in terms of their developmental requirements [19]. Uncertainty surrounding model parameter values could be communicated by combining or comparing outputs of multiple models for a species or via a sensitivity analysis [15,90,92]. 

Phenology models should somehow, but seldom do, include estimates of monitoring or sampling errors [93]. The best that can often be accomplished is to rely on observations collected by researchers or that have been verified by multiple people, such as “Research Grade” observations in the iNaturalist database [https://www.inaturalist.org (accessed on 19 December 2023)]. However, there is no guarantee that sampling errors can be minimized even when using published or verified observations [94]. For example, many insect monitoring studies report weekly trapping data (i.e., the trap is visited once per week), and therefore precise dates of phenological events are unknown. The observation data could be assigned as the date of data collection or the mid-date of the week-long interval; however, both approaches may result in imprecision of up to ±7 days. A 7-day error in phenological predictions may be considered adequate for most management models [90].

It should be noted, however, that model and sampling errors are often lower than errors resulting from intrinsic population variability. Many species populations, for reasons including selection pressures due to climate variability in time and space, are spread widely phenologically [23,95]. Sometimes, this is referred to as seasonal plasticity [96]. Many exhibit bimodal or trimodal behaviors in order to “hedge their bets” in uncertain environments. This phenological heterogeneity can result in wide errors and uncertainties in sampling and unexpected discrepancies in model predictions. Non-gaussian population spreads, such as bimodal flight distributions, are best described through larger sampling efforts and through the full characterization of such distributions, rather than using means and standard deviations for such populations. Even then, small outlier percentages in populations that happen to get sampled can sometimes be used to discredit or disprove otherwise valid models. 

## 7. Increasing Model Realism While Maintaining Simplicity

A common question in phenology modeling discussions is whether to use a simple model that is adaptable to multiple species versus a more complex, single-species model that can potentially deliver greater realism and accuracy [17,49,50,52,55]. Our current assessment is that using simple linear degree-day accumulations to display either developmental stages (via lookup table) or dates of phenological events are often adequate in their predictive accuracy, while still simple enough to be readily adapted for dozens, or even hundreds, of species. For example, validation analyses of certain models used by the DDRP, Pheno Forecast, and SAFARIS platforms have revealed evidence of overall good predictive performances [15,16,46,47]. To our knowledge, degree-day maps based upon nonlinear temperature response rates have not reached implementation for tactical decision support, despite the trend of using nonlinear equations to model the temperature–development response of insects [24,49,52,67,97].

A population-based approach to phenological mapping may improve predictive accuracy because it helps account for developmental variation that naturally occurs within insect populations [98,99,100,101,102,103]. DDRP is a population modeling platform [16] that is most similar to the still relevant and still in use “grandfather” of phenology modeling platforms, Predictive Extension Timing Estimator (PETE) [8,27]. Both platforms use a cohort approach to population modeling that involves tracking the development of population cohorts through all life stages over the year using a daily time step. Cohorts may start development at different times, which produces a distribution of times in which they transition into a new life stage (e.g., egg to larva) or undergo a particular phenological event [99,103]. One shortcoming of the PETE models is that they allow for only a single set of developmental thresholds for all life stages, plus an additional separate temperature threshold parameter for mating and oviposition. Conversely, DDRP allows for separate thresholds for each life stage, although most models developed to date have not taken advantage of this feature.

For many, if not most, insects in temperate climates, the use of single-factor degree-day models (i.e., temperature) have proven adequate as evidenced by their successful implementation for many agricultural pest species [9,10,15,20]. However, multifactor models that include additional driving variables, such as chilling, day length, and water availability, may be appropriate for certain species [48,61,70,73,92,104,105,106,107,108,109]. In seasonal environments, insects may enter and/or exit diapause in response to day length (photoperiod) to align their life cycles with favorable environmental conditions and resources [102,110,111,112]. Moisture may influence insect development by acting as a stimulus for diapause, modulator of developmental or reproductive rates, or behavioral cue for vital seasonal events [108,109,112,113]. For some species, a winter chill requirement may be required prior to degree-day driven development [110]. 

As an example, the brown marmorated stink bug, an important pest that has recently invaded and spread across the United States, has a reproductive diapause that terminates in mid-spring in response to photoperiod [114]. Modeling of this pest would likely be improved by using a two-phases approach in which photoperiod-cued diapause termination is followed by a degree-day response phase for spring and summer development [106,107]. A similar approach may be appropriate for other invasive insects, such as the Colorado potato beetle [*Leptinotarsa decemlineata* (Say)] [70], box tree moth [*Cydalima perspectalis* (Walker)] [92], European cherry fruit fly [*Rhagoletis cerasi* (L.)] [115], spotted-wing drosophila [*Drosophila suzukii* (Matsumura)] [116], and Asian tiger mosquito [*Aedes albopictus* (Skuse)] [117]. 

Few studies to date have included non-temperature factors in spatial phenology models for insects. However, spatial models for the biological control insects, *G. calmariensis* and Japanese knotweed psyllid [*Aphalara itadori* (Shinji)], incorporated photoperiodism to more accurately model voltinism in these species, both of which have photoperiodic-sensitive life stages that enter diapause in response to shortening day lengths [61,105]. These models calculated the day length for each latitude and day of year using the method of Forsythe et al. [118]. For certain insect pests, the inclusion of precipitation data in site-based phenology models has improved predictive accuracy of the timing of management-relevant phenological events, such as the first adult emergence of the saddle gall midge *Haplodiplosis marginata* (von Roser) [104] and flight activities of pest aphids [72]. However, to our knowledge, moisture factors have not yet been included in spatial phenology models for insects, despite the availability of gridded daily precipitation data in certain data products (Table 3). 

Models that account for geographic variation in life history traits may be necessary to accurately predict phenology of insects in new environments [23,52,105,119]. For example, Grevstad et al. [105] used different values for the critical day length parameter in spatialized phenology models for the Japanese knotweed psyllid owing to genetically based differences in this species’ critical photoperiod in the native range. However, defining optimal parameter values would create the need for laboratory studies of insects sampled from across the species’ distribution, which may be infeasible. Even if these data were available for the entire known distribution, they may not be useful for newly introduced species that have unknown geographic origins [61]. Therefore, it is most prudent, with this lack of full understanding of the life cycle, to model what is understood and simply state the model’s shortcomings so that they can be considered by managers.

Accurately mapping the phenology of migratory species may require the development of custom models that include multiple abiotic and biotic variables [76,77]. Real-time phenological mapping platforms for CONUS include models for several migratory species, including the old world bollworm [(Helicoverpa armigera (Hübner)], silver Y moth [*Autographa gamma* (L.)], and Sunn pest (*Eurygaster integriceps* Puton) (Table 2). Assessing the predictive accuracy of these models for CONUS is not yet possible because the pests are not established in this region. However, incorporating datasets, such as atmospheric dispersion models and host plant phenology, into models for migratory insects could potentially improve the predictive accuracy [76,77]. 

## 8. Model Validation

Errors in predictions of phenology models should be estimated when validation data are available [20,55,90]. Model overprediction, in which events or life stages are predicted later than observed dates, is typically more problematic than underprediction because decision-makers may miss the best opportunity to detect or manage populations. Communicating potential model errors to end users of phenological maps may allow them to adjust the timing of their surveillance and management activities accordingly [55]. Similarly, researchers should clearly communicate that a model is presumptive if validation analyses are infeasible or insufficient owing to an absence or paucity of data. In general, clear communication of potential errors and uncertainties in models is important for creating and maintaining end users’ trust and increasing the likelihood that they use forecasts for decision support [3,11,94].

Phenology model validation requires a set of observations not used in model development. Observations could be resampled using jackknifing or cross-validation methods, split into separate sets (e.g., 75% for development, 25% for validation), or originate from different years or areas [31,87,90,120,121]. Potential sources of data for model validation include peer-reviewed literature, reports, graduate theses, unpublished monitoring studies, and online databases such as iNaturalist, Nature’s Notebook [https://www.usanpn.org/natures_notebook (accessed on 19 December 2023)], and the National Agricultural Pest Information System [https://cmr.earthdata.nasa.gov/search/concepts/C1214608223-SCIOPS (accessed on 19 December 2023)] [15,16,46,47]. Caution should be taken when using citizen science collected data (e.g., from iNaturalist) because they may contain errors, such as species misidentifications or incorrect dates and locations [93,94].

Validating a spatial phenology involves extracting gridded model predictions for each georeferenced observation using geographic information system (GIS) software. Ideally, the geographic precision of a phenological observation should be equal to or greater than the spatial resolution of climatic datasets used for modeling. For CONUS, this would correspond to spatial resolutions of 800 m to 4 km depending on the climatic dataset (Table 3). If observations lack coordinate data but geographic information, such as a city name, is known, then one could calculate the average of predicted values within the area delineated by geospatial data, such as cartographic boundaries of cities [16,46]. However, this solution is likely problematic for areas where phenology may vary substantially over short distances, such as a large or topographically complex county.

The statistics for phenological model validation, keeping in line with the principle of parsimony, should be done using readily understood terms. Summary statistics may include the mean difference between predicted and observed dates (day of year, DOY), along with variability estimates, such as 95% confidence intervals, to provide an estimate of the overall error including the bias [55,87,90]. The bias may be estimated as the average amount by which predicted DOYs are greater than observed DOYs, whereas the mean absolute error (i.e., the average absolute difference) can estimate the expected number of days a given prediction might be in error. 

Model performance may also be evaluated using statistical tests, such as confusion matrices or equivalence tests [46,47,90]. A confusion matrix indicates the rate of true versus false positives and true versus false negatives, which allows model sensitivity, specificity, and accuracy of model predictions to be calculated. For example, confusion matrices were used to evaluate Pheno Forecast maps for the hemlock wooly adelgid, *Adelges tsugae* (Annand), in CONUS [47]. Conversely, an equivalence test may involve testing the null hypothesis that the means of predicted and observed DOYs differ by a specified equivalence interval (i.e., number of days). This method was used to evaluate predictions of adult activity produced by the DDRP model for the emerald ash borer [46]. A *t*-test (comparison of means), F-test (comparison of standard deviation), and Kolmogorov–Smirnov test (computing the maximum distance between the cumulative distribution of two samples) are other potential options for phenology model validation [90].

## 9. Recommendations for Future Research

Usability tests that allow end users to compare and evaluate different types and formats of phenological maps for invasive insects may help increase map uptake for decision support [47]. For example, usability tests are needed to assess preferences for degree-day lookup table maps vs. phenological event maps, which are relatively new and untested with regard to end user acceptance. As the emerald ash borer adult emergence and spongy moth egg hatch is of operational interest in areas of current expansion, these might serve as potential target subject areas to perform these tests. Also, the release of open-source code for phenological mapping platforms, as has been done with DDRP, may help increase the development and deployment of phenological event maps, which should further enhance end user acceptance.

The development and delivery of real-time gridded daily *T_min_* and *T_max_* datasets for regions beyond CONUS is essential for expanding the use of phenological maps for within-season decision support. The RTMA/UTMA products from the National Oceanic and Atmospheric Administration (NOAA) include predictions for the entire United States, Puerto Rico, and Guam, and they have acceptable temperature error rates for some needs, such as for air traffic control [122]. An area needing further investigation is whether quasi-global temperature datasets, such as the ERA5-Land hourly temperature dataset [123], are sufficient for phenological mapping, at least for historical time periods, owing to delays in data releases (e.g., a 2–3 months for ERA5-Land data). 

The number of models developed for real-time phenological mapping is extremely low (Table 2) when considering that over 10,000 invasive alien insect species exist worldwide, many of which are intercepted in ports of entry each year [124,125,126,127]. Future modeling work could focus on species that represent the highest risk to agriculture and natural resources, such as those on the USDA APHIS PPQ’s “National Priority Pest List” for the United States [https://approvedmethods.ceris.purdue.edu (accessed on 20 December 2023)]. Spatial phenology models for pests that are small, cryptic, or occur in cryptic habitats, such as underground or inside of trees during parts of their life cycle, may be particularly useful for early detection efforts [46,47]. Species with known temperature thresholds and developmental requirements could also be targeted for model development [19,48]. 

The development of multifactor phenology modeling platforms will help increase the number of insect pests that may be accurately modeled. For example, the incorporation of precipitation or soil moisture factors in the phenology modeling process could potentially improve the predictive accuracy for species that use soil moisture as a cue for reproduction or diapause termination [108,109,128]. Real-time gridded daily precipitation data for CONUS are available in the PRISM database. Additionally, hourly estimates of soil moisture in the National Aeronautics and Space Administration’s (NASA) Soil Moisture Active Passive satellite product series [https://smap.jpl.nasa.gov/data (accessed on 20 December 2023)] may be useful for modeling moisture-sensitive insect species. Predicting host plant phenology in real-time will be facilitated by the development of remotely sensed vegetation phenology datasets at high spatial resolutions, such as those derived from real-time MODIS data [https://www.earthdata.nasa.gov/learn/find-data/near-real-time/modis (accessed on 20 December 2023)] [129,130]. Future work could also explore the possibility of including atmospheric dispersion models, such as NOAA’s Hybrid Single-Particle Lagrangian Integrated Trajectory model (HYSPLIT) in models for migratory insect pests [76,77,131].

Phenological maps will be important for assessing the impacts of climate change on pest phenology. Warming temperatures, an increase in frequency and severity of extreme weather events (e.g., heat waves and droughts), and altered precipitation patterns under global warming are expected to impact pest phenology as well as phenological synchrony between pests and their host plants [23,34,69,72,74,132,133,134]. Shorter, warmer winters in many regions will likely promote overwintering survival and increase developmental rates of invasive insects, allowing them to emerge earlier and attain higher densities over a longer growing season [42,68,69,71,92,117]. For migratory insects, a trend towards earlier flight activity is expected to change the timing and severity of pest outbreaks [72,135]. Estimating temporal trends in voltinism and phenological events of pests and their host plants may help identify regions at greatest risk of being invaded or experiencing high pest pressure in the future [34,42,46,68,69,70,71,135]. Additionally, this information may help with developing long-term surveillance and management strategies [15]. 

## 10. Conclusions

Phenological maps can provide insight into the development and seasonal activities of invasive insects at area-wide scales, such as counties, states, or entire nations. Several web-based platforms offer generic degree-day maps, degree-day lookup table maps, and phenological event maps to support within-season decision-making for the detection and control of invasive insects for CONUS. With the development of real-time gridded climate datasets for more regions of the world, phenological maps could become more commonly used for planning pest surveillance and management activities. Phenological maps may also be used to assess establishment risk and to investigate pest–host interactions and climate-driven changes in pest phenology. We encourage modelers to quantify and communicate model error and uncertainty whenever possible and to engage with map end users to improve and promote products. 

## Figures and Tables

**Figure 1 insects-15-00006-f001:**
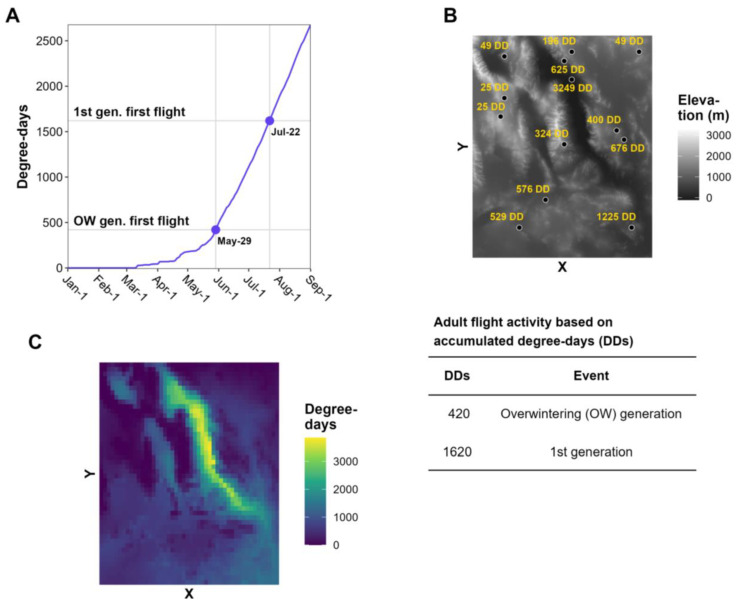
Degree-day (DD) accumulations for 1 January to 1 September predicted by site-based and spatial phenology models. (**A**) Plot of date vs. DD accumulations depicts dates of adult flight activity at a single site. (**B**) An elevation map with predictions for multiple sites is shown with a key of DD requirements for adult flight activity. (**C**) Phenological map of the same area as (**B**) shows continuous predictions. The X and Y axis in (**B**,**C**) indicate longitude and latitude, respectively.

**Figure 2 insects-15-00006-f002:**
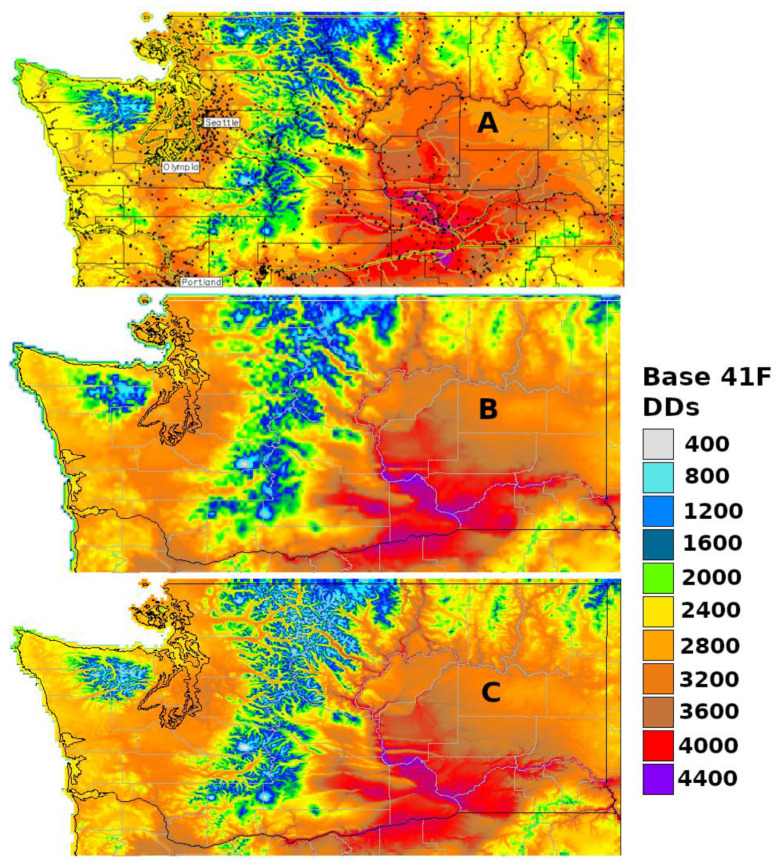
Example cumulative degree-day (DD) mapping products for Washington state generated online at USPest.org [https://uspest.org (accessed on 19 December 2023)]. All maps depict DD accumulation [base 41 °F (5 °C)] between 1 January and 31 August of 2020 (calculated using the single triangle method). (**A**) Map generated using an older version of the program [https://uspest.org/cgi-bin/usmapmaker.pl (accessed on 19 December 2023)] Awith a spatial resolution of 800 m. Black dots indicate the locations of weather stations used in the correction of monthly PRISM-based cumulative DDs. (**B**) Map produced by the newer program [https://uspest.org/dd/mapper (accessed on 19 December 2023)] that uses daily PRISM data with a coarser (4 km) spatial resolution (no weather station correction required). (**C**) Downscaled output produced using the newer program (800 m resolution).

**Figure 3 insects-15-00006-f003:**
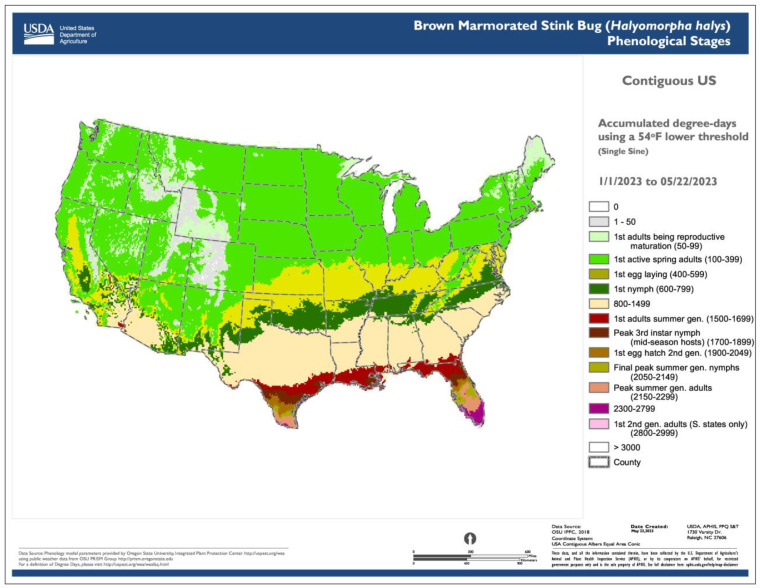
Degree-day lookup table map for the brown marmorated stink bug [*Halyomorpha halys* (Stål)] produced as part of the Spatial Analytic Framework for Advanced Risk Information Systems (SAFARIS) Field Operations Weekly map series [59]. The map uses a degree-day lookup table to associate cumulative degree-days with predicted life stages present across the contiguous United States on 22 May 2023. Thus, it provides a “snapshot in time” of phenology model predictions for a specific date. Reproduced with permission from SAFARIS, Brown Marmorated Stink Bug (*Halymorpha halys*) Phenological Stages. Published by SAFARIS, U.S. Department of Agriculture (USDA), and North Carolina State University, 2023. Available online: https://safaris.cipm.info (accessed on 19 December 2023).

**Figure 6 insects-15-00006-f006:**
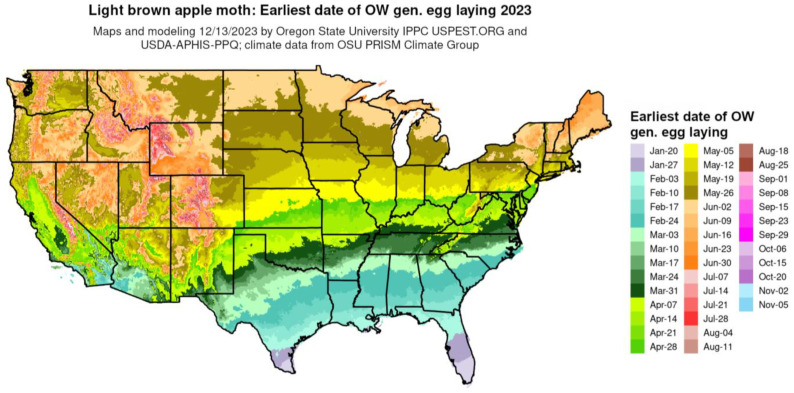
Phenological event map depicting the earliest date of egg laying by the overwintered (OW) generation of the light brown apple moth (*Euphyas postvittana*) for 2023 produced by DDRP (map issue date: 13 December 2023).

**Table 1 insects-15-00006-t001:** Presently available web-based platforms that produce generic degree-day (DD) maps, DD lookup table maps, and/or phenological event maps for invasive insect species in the contiguous United States. All platforms are publicly available and accessible via a web browser.

Map Types	Platform	Organization	Website
Generic DD	AgWeather Vegetable Disease & Insect Forecasting Network	UW	https://agweather.cals.wisc.edu/vdifn?p=insect (accessed on 19 December 2023).
Generic DD	Enviroweather	MSU	https://www.enviroweather.msu.edu/ (accessed on 19 December 2023).
Generic DD	USPest degree-day mapping	OSU	https://uspest.org/cgi-bin/usmapmaker.pl (accessed on 19 December 2023).
DD lookup table, phenological events	DDRP	OSU	http://uspest.org/CAPS/ (accessed on 19 December 2023).
DD lookup table	PestCAST and FOWeekly	SAFARIS	https://safaris.cipm.info (accessed on 19 December 2023).
Phenological events	Pheno Forecasts	USA NPN	https://www.usanpn.org/news/forecasts (accessed on 19 December 2023).

UW = University of Wisconsin, MSU = Michigan State University, OSU = Oregon State University, SAFARIS = Spatial Analytic Framework for Advanced Risk Information Systems, USA NPN = USA National Phenology Network.

**Table 2 insects-15-00006-t002:** Invasive insect species for which spatial degree-day models are available for the contiguous United States. Pest establishment status (yes/no) and map series name(s) are indicated for each species.

Species	Established	Series
Alfalfa weevil [*Hypera postica* (Gyllenhal)]	Yes	FO Weekly
Asian longhorn beetle [*Anoplophora glabripennis* Motschulsky)]	Yes	DDRP, FO Weekly, PestCAST, Pheno Forecast
Asiatic rice borer (*Chilo suppressalis* Walker)	No	DDRP
Black spruce beetle [*Tetropium castaneum* (L.)]	No	FO Weekly
Brown marmorated stinkbug [*Halymorpha halys* (Stål)]	Yes	FO Weekly
Brown spruce longhorn beetle [*Tetropium fuscum* (F.)]	No	FO Weekly
Box tree moth [*Cydalima perspectalis* (Walker)]	Yes	FO Weekly, PestCAST
Cereal leaf beetle [*Oulema melanopus* (L.)]	Yes	FO Weekly
Cotton cutworm [*Spodoptera litura* (F.)]	No	DDRP, FO Weekly
Egyptian cottonworm [*Spodoptera littoralis* (Boisduval)]	No	DDRP
Emerald ash borer (*Agrilus planipennis* Fairmaire)	Yes	DDRP, FO Weekly, Pheno Forecast
European cherry fruit fly [*Rhagoletis cerasi* (L.)]	Yes	PestCAST
European grapevine moth [*Lobesia botrana* (Denis & Schiffermüller)	No	FO Weekly
False codling moth [*Thaumatotibia leucotreta* (Meyrick)]	No	DDRP
Hemlock woolly adelgid [*Adelges tsugae* (Annand)]	Yes	Pheno Forecast
Honeydew moth (*Cryptoblabes gnidiella* Millière)	No	FO Weekly, DDRP
Japanese beetle (*Popillia japonica* Newman)	Yes	FO Weekly, PestCAST
Japanese flower thrips (*Thrips setosus* Moulton)	No	FO Weekly
Japanese pine sawyer beetle (*Monochamis alternatus* Hope)	No	DDRP
Light brown apple moth [*Epiphyas postvittana* (Walker)]	Yes	FO Weekly, DDRP
Oak ambrosia beetle [*Platypus quercivorus* (Murayama)]	No	DDRP
Old world bollworm [*Helicoverpa armigera* (Hübner)]	No	DDRP, FO Weekly, PestCAST
Pine tree lappet moth [*Dendrolimus pini* (L.)]	No	DDRP
Pink bollworm [*Pectinophora gossypiella* (Saunders)]	Yes	FO Weekly
Silver Y moth [*Autographa gamma* (L.)]	No	FO Weekly, DDRP
Sirex woodwasp [*Sirex noctilio* (F.)]	Yes	FO Weekly
Small tomato borer [*Neoleucinodes elegantalis* (Guenée)]	No	DDRP
Spongy moth [*Lymantria dispar* (L.)]	Yes	FO Weekly, PestCAST, Pheno Forecast
Spotted lanternfly [*Lycorma delicatula* (White)]	Yes	FO Weekly, PestCAST
Summer fruit tortrix [*Adoxophyes orana* (Fischer von Rösslerstamm)]	No	FO Weekly
Sunn pest (*Eurygaster integriceps* Puton)	No	DDRP
Tomato leafminer [*Tuta absoluta* (Meyrick)]	No	DDRP
Winter moth [*Operophtera brumata* (L.)]	Yes	Pheno Forecast

**Table 3 insects-15-00006-t003:** Gridded daily *T_min_* and *T_max_* datasets that may be suitable for phenological mapping. The temporal coverage, spatial resolution, developer or provider, and URL for each data product are provided. 0.1° = ca. 11.1 km at the equator.

Region	Product	Temporal Coverage	Spatial Resolution	Developer	URL
Global	NMME	Forecasts up to 12 months	0.1°	National Oceanic and Atmospheric Administration	https://www.cpc.ncep.noaa.gov/products/NMME (accessed on 19 December 2023).
CONUS	PRISM	1981 to present	4 km, 800 m	PRISM Climate Group, Oregon State University	http://prism.oregonstate.edu (accessed on 19 December 2023).
CONUS, HI, GU, PR, AK	RTMA	2019 to present	2.5 km (CONUS, HI, GU), 3 km (AK), 1.25 km (PR)	National Oceanic and Atmospheric Administration	https://www.nco.ncep.noaa.gov/pmb/products/rtma (accessed on 19 December 2023).
CONUS	TopoWX	1948 to 2016	800 m	University of Montana	http://www.ntsg.umt.edu/project/topowx.php (accessed on 19 December 2023).
CONUS	METDATA	1979 to present	4 km	University of Idaho	https://www.sciencebase.gov/catalog/item/54dd5df2e4b08de9379b38d8 (accessed on 19 December 2023).
CONUS, HI, GU, PR, VI, AK, NPOI	NDFD	Forecasts up to 7 days	5 km (CONUS), 2.5 km (HI, GU), 1.25 km (PR, VI), 6 km (AK), 10 km (NPOI)	National Oceanic and Atmospheric Administration	https://vlab.noaa.gov/web/mdl/ndfd (accessed on 19 December 2023).
Europe	E-OBS	1950 to previous month	0.1°	EU-FP6 project UERRA (Uncertainties in Ensembles of Regional ReAnalyses)	https://surfobs.climate.copernicus.eu (accessed on 19 December 2023).
North America	Daymet	1980 to previous calendar year	1 km	Oak Ridge National Laboratory, University of Montana	http://www.ntsg.umt.edu/project/daymet.php (accessed on 19 December 2023).
Bangladesh, Nepal, and Pakistan	Unknown	1981 to 2016	5 km	Ali et al. [78]	https://doi.org/10.6084/m9.figshare.21565149.v1 (accessed on 19 December 2023).
Brazil	BR-DWGD	1980 to 2016	0.1°	Xavier et al. [79]	https://sites.google.com/site/alexandrecandidoxavierufes/brazilian-daily-weather-gridded-data (accessed on 19 December 2023).
China	CDAT	1999 to 2018	0.1°	Fang et al. [80]	https://zenodo.org/record/5513811#.YtnLNXbMIuU (accessed on 19 December 2023).
China	HRLT	1961 to 2019	0.1°	Qin et al. [81]	https://doi.org/10.1594/PANGAEA.941329 (accessed on 19 December 2023).
India	HRDGT	1951–2016	0.1°	Nengzouzam et al. [82], India Meteorological Department	https://searchworks.stanford.edu/view/13160050 (accessed on 19 December 2023).
UK	HadUK-Grid	1884 to present	1 km	Met Office	https://www.metoffice.gov.uk/research/climate/maps-and-data/data/haduk-grid/haduk-grid (accessed on 19 December 2023).

CONUS = contiguous United States, HI = Hawaii, GU = Guam, PR = Puerto Rico, VI = Virgin Islands, AK = Alaska, NPOI = North Pacific Ocean Islands.

## Data Availability

The data presented in this study are available on request from the first author or corresponding author.

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
