# Peer review of "Phenological Mapping of Invasive Insects: Decision Support for Surveillance and Management"

_insects, 2023, doi:10.3390/insects15010006_

Round 1

Reviewer 1 Report

Comments and Suggestions for Authors

It is a very good paper. I have only a few comments and recommendations, but it is mostly up to the authors to follow or reject (except to Latin names issue).

Author Response

Dear Reviewer,

Thank you very much for your helpful comments and suggested edits. Our responses to your comments are included in the attached PDF. 

Best wishes,

Brittany

Reviewer 2 Report

Comments and Suggestions for Authors

This review article is very well written and provides an interesting and useful review of tools available for mapping potential invasive insect phenology in space and time. I’m not familiar with all of the options, so I can’t evaluate this manuscript for completeness in that sense.

My only suggestion for improving this manuscript would be to include a discussion of insect migration as an element in understanding and predicting insect outbreaks. Several of the species listed in Table 2 are migratory. Predictive phenology may depend on degree day measurements on a continental basis combined with atmospheric modeling of wind directions at high altitudes. There is very little information available about which species are migratory in which regions, with a few exceptions for species with global implications, such as fall armyworm (Spodoptera frugiperda). Still, I feel that a review like this manuscript would benefit from at least acknowledging the implications of long-distance migration and directly suggesting the topic for future research. I'll provide a few possible citations:

Lv, H., M.-Y. Zhai, J. Zeng, Y.-Y. Zhang, F. Zhu, H.-M. Shen, K. Qiu, B.-Y. Gao, D. R. Reynolds, J. W. Chapman and G. Hu (2023). "Changing patterns of the East Asian monsoon drive shifts in migration and abundance of a globally important rice pest." Global Change Biology https://doi.org/10.1111/gcb.16636

Tay, W. T., R. L. Meagher, C. Czepak and A. T. Groot (2023). "Spodoptera frugiperda: Ecology, Evolution, and Management Options of an Invasive Species." Annual Review of Entomology 68(1): 299-317.

Wang, H. H., W. E. Grant, N. C. Elliott, M. J. Brewer, T. E. Koralewski, J. K. Westbrook, T. M. Alves and G. A. Sword (2019). "Integrated modelling of the life cycle and aeroecology of wind-borne pests in temporally-variable spatially-heterogeneous environment." Ecological Modelling 399: 23-38.

Westbrook, J., S. Fleischer, S. Jairam, R. Meagher and R. Nagoshi (2019). "Multigenerational migration of fall armyworm, a pest insect." Ecosphere 10(11) DOI: 10.1002/ecs2.2919

Westbrook, J. K. and J. D. Lopez (2010). "Long-Distance Migration in Helicoverpa zea: What We Know and Need to Know." Southwestern Entomologist 35(3): 355-360.

Zeng, J., Y. Q. Liu, H. W. Zhang, J. Liu, Y. Y. Jiang, K. A. G. Wyckhuys and K. M. Wu (2020). "Global warming modifies long-distance migration of an agricultural insect pest." JOURNAL OF PEST SCIENCE 93(2): 569-581.

Author Response

Dear Reviewer,

Thank you very much for your helpful suggestions regarding migratory insect pests. Please see that attached document for our responses.

Best wishes,

Brittany

Reviewer 3 Report

Comments and Suggestions for Authors

MS is well organized and structured showing examples for the established and newly introduced pests. 

Author Response

Thank you for your positive feedback!